Genome-wide identification and analysis of Lateral Organ Boundaries Domain (LBD) transcription factor gene family in melon (Cucumis melo L.)

Derelli Tufekci Ebru ederelli@karatekin.edu.tr
Department of Field Crops, Food and Agriculture Vocational High School, Cankiri Karatekin University , Cankiri , Turkey
Sotelo-Mundo Rogerio
Electronic publication date: 2023 Sep 29
Publication date: 2023
Volume: 11
Electronic Location ID: e16020
Received 2023 May 17; Accepted 2023 Aug 11
Copyright: ©2023 Derelli Tufekci
Copyright year: 2023
Copyright holder: Derelli Tufekci
License: This is an open access article distributed under the terms of the Creative Commons Attribution License, which permits unrestricted use, distribution, reproduction and adaptation in any medium and for any purpose provided that it is properly attributed. For attribution, the original author(s), title, publication source (PeerJ) and either DOI or URL of the article must be cited.
License URL: https://creativecommons.org/licenses/by/4.0/

Keywords: Cucumis melo, LBD transcription factor, Gene expression, Evolution

Funding: Cankiri Karatekin University Scientific Research Projects Coordinator (BAP) SMYO260722B15 This study was financially supported by the Cankiri Karatekin University Scientific Research Projects Coordinator (BAP) under the project number SMYO260722B15. The funders had no role in study design, data collection and analysis, decision to publish, or preparation of the manuscript.

==============================
Background

Lateral Organ Boundaries Domain (LBD) transcription factor (TF) gene family members play very critical roles in several biological processes like plant-spesific development and growth process, tissue regeneration, different biotic and abiotic stress responses in plant tissues and organs. The LBD genes have been analyzed in various species. Melon (Cucumis melo L.), a member of the Cucurbitaceae family, is economically important and contains important molecules for nutrition and human health such as vitamins A and C, β-carotenes, phenolic acids, phenolic acids, minerals and folic acid. However, no studies have been reported so far about LBD genes in melon hence this is the first study for LBD genes in this plant.

Results

In this study, 40 melon CmLBD TF genes were identified, which were separated into seven groups through phylogenetic analysis. Cis-acting elements showed that these genes were associated with plant growth and development, phytohormone and abiotic stress responses. Gene Ontology (GO) analysis revealed that of CmLBD genes especially function in regulation and developmental processes. The in silico and qRT-PCR expression patterns demonstrated that CmLBD01 and CmLBD18 are highly expressed in root and leaf tissues, CmLBD03 and CmLBD14 displayed a high expression in male-female flower and ovary tissues.

Conclusions

These results may provide important contributions for future research on the functional characterization of the melon LBD gene family and the outputs of this study can provide information about the evolution and characteristics of melon LBD gene family for next studies.

Introduction

In order to active transcription, transcription factors (TFs) bind to the promoter or enhancer of a gene and and have specific DNA-binding sites. TFs play crucial roles in responding to stress regulating the cell cycle, transmitting signals between cells, and controlling the growth and development of plants (Mahajan & Tuteja, 2005). Among TFs, the Lateral Organ Boundaries Domain (LBD) gene family, which encodes proteins containing plant-specific lateral organ boundaries (LOB) domains, also called Asymmetric Leaves2-Like (ASL) gene family, is the TF class found only in higher plants (Xu, Luo & Hochholdinger, 2016).

The gene family is located at the base of the primary lateral organ and was initially identified in Arabidopsis thaliana using enhancer traps (Shuai, Reynaga-Pena & Springer, 2002). LBD TFs contain three unique protected structures organized from the N to the C terminus: the zinc finger-like C-block (CX2CX6CX3C), the Gly-Ala-Ser-block (GASblock), and the leucine-like zipper module (LX6LX3LX6L). C block is for DNA binding, the GAS block plays a role in the function of LBD proteins. This motif is located in the center of the LOB structural domain (Majer & Hochholdinger, 2011). The LBD TF genes are divided into subclasses Class I and Class II according to their structural features. Whereas Class I can be grouped into Ia, Ib, Ic, and Ie which contain all the modules, Class II can be grouped into IIa and IIb, which contain only the conserved zinc finger-like structural domain (Matsumura et al., 2010). After the first identification of the LBD gene family in A. thaliana (Yang, Yu & Wu, 2006), they have been identified in many other plant species, such as Solanum lycopersicum, Zea mays, Vitis vinifera, Malus domestica, Glycine max, Morus notabilis (Wang et al., 2013a; Wang et al., 2013b; Zhang, Zhang & Zheng, 2014; Cao et al., 2016; Luo et al., 2016; Yang et al., 2017). Studies have shown that the LBD TF genes play key roles in various biological processes like plant-specific development and growth process, tissue regeneration, and response to different biotic-abiotic stresses in plant tissues and organs (Majer & Hochholdinger, 2011). In addition, LBD genes participated in phytohormone accumulation, nitrogen metabolism (Bell et al., 2012).

Previous studies revealed that AtLOB/AtASL4 is specifically expressed in the base of the lateral organ proximal axis of in A. thaliana (Shuai, Reynaga-Pena & Springer, 2002). Cytokinin-regulated AtLBD3/AtASL9 takes part in the regulation of plant development, while AtLBD6/AtAS2 regulates KNOX gene expression and inhibits cell proliferation (Iwakawa et al., 2007) in A. thaliana. AtLBD15 which is regulated by Wuschel (WUS) gene was shown to participate in apical meristem cell differentiation (Sun et al., 2013). AtLBD16 and AtLBD18 involve in the initiation and occurance of A. thaliana lateral roots (Lee et al., 2009). Additionally, it was reported that OsLBD37 and OsLBD38 highly expressed in rice heading and increased yields in Oryza sativa (Li et al., 2017a; Li et al., 2017b).

As a member of the Cucurbitaceae family, melon (Cucumis melo L.) is an annual species. It has a 480 million base pairs genome and 12 chromosomes (2n = 24). Its genome was first published in 2012 (Garcia-Mas et al., 2012). Besides being economically significant, it synthesizes A and C vitamins, β-carotenes, phenolic acids and minerals, and folic acid, which are significant in terms of nutrition and human health (Wu et al., 2020).

Research on the LBD genes has been conducted on numerous species, but there has been no data reported on the characteristics of LBD genes in melons. In this study, 40 CmLBD TF genes in melons, were identified and analyzed using bioinformatics tools to determine their chromosome location, conserved domain features, genetic structures, evolutionary relationships, and expression.

Materials & Methods

Identifcation and characterization of the melon CmLBD genes

The C. melo (DHL92 v3.5.1) genome v3.6.1 was downloaded from the Cucurbit Genomics Database (CuGenDB) (http://cucurbitgenomics.org) to identify the LBD TF gene family members. Additionally, whole genome data for A. thaliana and Cucumis sativus (Cucumber) were obtained from Arabidopsis Information Resource (TAIR10) database (http://www.arabidopsis.org/) and the Cucurbit Genomics Database (http://cucurbitgenomics.org/), respectively. These genomes were used for Blast in Phytozome database v13. Hidden Markov model of LOB domain (DUF260, PF03195) information was retrieved from the Pfam database (http://pfam.xfam.org/) and used for an HMMER3 software of the local melon protein database (E ≤10−20) (Johnson, Eddy & Portugaly, 2010). Open reading frame (ORF) length, molecular weight (MW), isoelectric point (pI), grand average of hydropathicity (GRAVY) of CmLBD members were analyzed by online tool ExPASy (http://web.expasy.org/protparam/) and the subcellular localizations of the LBD genes was determined with Cell-PLoc 2.0 (http://www.csbio.sjtu.edu.cn/bioinf/Cell-PLoc-2/).

Chromosomal location and phylogeny analysis of the CmLBD gene family in melon

The chromosome localization of LBD TF gene members was mapped unto melon chromosomes using the online tool MapGene2Chromosome-MG2C (http://mg2c.iask.in/mg2c_v2.1/, (accessed on 20 January 2023)).

43 A. thaliana LBD proteins and 39 cucumber LBD proteins were defined from HMMER3 searches of their respective local protein databases (Finn, Clements & Eddy, 2011). The CmLBD protein sequences were aligned by the ClustalW program in MEGA7. The Neighbor-Joining method (NJ) in the MEGA7 program was utilized to construct a phylogenetic tree (Thompson et al., 1997). The bootstrap value was 1000. LBD members in Arabidopsis thaliana, Cucumis sativus and Cucumis melo were compared. The phylogenetic tree was visualized through IQ-TREE v2.0.3 (Minh et al., 2020) and FigTree v1.4.4 (http://tree.bio.ed.ac.uk/software/figtree/) software.

Analysis of CmLBD gene structure and protein conserved motifs

The Perl language program was used to obtain annotations of the LBD gene. Gene Structure Display Server v2 (GSDS: http://gsds.gao-lab.org/) was used to assign the exon–intron structure of the CmLBD genes (Hu et al., 2015). To predict the motifs Multiple Em for Motif Elicitation (MEME) web tool (https://meme-suite.org/meme/) and the program parameters defined by Bailey et al. (2006).

Synteny analysis

Protein sequences of the melon were aligned with each other or with the protein sequences from A. thaliana and cucumber using TBtools v1.108 software (Chen et al., 2020). The Multiple Collinearity Scan (MCScanX) tool was used to identify gene duplication events and syntenic relationships between LBD proteins. Circos and Dual Synteny Plot in TBtools software were conducted to visualize the results (Lescot et al., 2002; Wang et al., 2012). To selective pressure analysis calculate the synonymous rate (Ks), non-synonymous rate (Ka), and Ka/Ks ratio of each gene pair via KaKs_Calculator 2.0 tool (Wang et al., 2010).

Promoter analysis of CmLBD gene family

Cis-acting member analysis of the melon CmLBD gene family was performed throughout the PlantCARE database in 5′upstream gene regions containing approximately 1.5 kb of nucleotide sequences. PlantCARE (http://bioinformatics.psb.ugent.be/webtools/plantcare/html/) (Lescot et al., 2002) and the TBtools were applied to create a visualization of the gene structure.

Gene Ontology (GO) annotation

Gene ontology analysis of CmLBD protein sequences was performed via the Blast2GO server (http://www.blast2go.com) (Conesa et al., 2005) to detect the biological processes. For this aim, the datasets were processed by BlastP, mapping, and annotation algorithms respectively.

Expression analysis of CmLBDs based on RNA-Seq data

To determine the CmLBD gene expression patterns in melon tissues and fruits. RNA-seq libraries were retrieved from Sequence Read Archive (SRA) (https://www.ncbi.nlm.nih.gov/sra). The raw sequences of female flowers, male flowers, leaves, roots, stems, and ovaries were loaded previously to NCBI (https://www.ncbi.nlm.nih.gov/, under the project number PRJNA803327, and the transcriptome libraries, including fruit climacteric (C), growing (G), post-climacteric (P) and ripening (R) stages were obtained from NCBI SRA with project number PRJNA543288 (Tian et al., 2019). After sequence quality control, each library RNA-Seq reads was mapped to CmLBD gene sequences. In this study, gene expression values were used fragments per kilobase of transcript per million mapped (FPKM) algorithm (Mortazavi et al., 2008). Heat maps were drawn with ClustVis software (https://biit.cs.ut.ee/clustvis/) (Metsalu & Vilo, 2015).

Plant growth and qRT–PCR analysis of CmLBD genes

C. melo cv. ‘Kirkagac’ seeds were taken from Ankara University, Department of Horticulture in Ankara, Turkey. Firstly, the melon seeds were surface-sterilized in ethanol and sodium hypochlorite. The seeds that were sterilized were germinated in Petri dishes at room temperature. The seedlings were sterilized and germinated at room temperature and transferred to Murashige and Skoog (MS) medium under controlled conditions after 4 days. After about 20 days, the plants were grown in pots under greenhouse conditions. After 20 days, plants were transferred to pots and grown under controlled greenhouse conditions. Root, stem, leaf, female flower, male flower, and anthesis ovary tissue samples from 60-day-old seedlings were then harvested, with three independent biological replicates per tissue. After harvest, tissue samples were stored at −80 °C for use in RNA isolation. Total RNA was isolated from tissues with TRIzol reagent (Invitrogen, Carlsbad, CA, USA). Approximately 1 µg of RNA was used to synthesize the first-strand cDNAs using the SuperScript™ III First-Strand Synthesis System. The primers were designed using Primer3 v4.1.0 program, and the actin was used as a housekeeping gene in the qRT-PCR analysis (Table S1). qRT-PCR reactions performed on the CFX Connect™ Real-Time PCR Detection System using SYBR™ Green PCR Master Mix per the manufacturer’s instructions. Every qRT-PCR reaction (25 µL) included 12.5 µL of 2 × real-time PCR Mix (SYBR Green I), 0.5 µL of primer and suitably diluted cDNA as a template. qRT-PCR conditions were 95 °C for 20 s, followed by 40 cycles of 95 °C for 30 s, 54 °C for 20 s and 72 °C for 10 s. Three biological and technical replicates for every sample were performed in all qRT-PCR reactions. The 2−ΔΔCT method was used to analyze the data, and one-way ANOVA was used for calculating statistics (Livak & Schmittgen, 2001).

Results

Identification of melon CmLBD gene family members

In this study, 40 LBD TF genes were described. Concerning the position on the chromosome, CmLBD genes were named from CmLBD01 to CmLBD40. The MW and pI values of CmLBD proteins were detected using ExPASy (http://web.expasy.org/protparam/) (Table S2). Their pI ranged from 4.4516 (CmLBD28) to 9.43 (CmLBD23). The ORFs ranged from 219 (CmLBD34) to 1089 (CmLBD16) bp, and Cell-Ploc subcellular localization prediction showed that all CmLBD genes are distributed in the nucleus. The CmLBD16 was the maximal protein with 362 amino acids and 39.74 KDa MW. The minimum protein was be CmLBD34, which has 72 amino acids and 8.13 KDa MW.

Phylogenetic relationships and gene structure analysis of the melon CmLBD gene family

The LBD proteins of 43 Arabidopsis (ATs), 39 cucumbers (KGNs) and 40 melon (CmLBDs) neighbor-joining with aligned amino acid sequences to build a phylogenetic tree (Fig. 1). The results showed that 122 CmLBD genes could be classified into two major groups: Class I and Class II. The larger group (Class I) was further divided into five sub-groups (Class Ia–Class Ie) and Class II was divided into two sub-groups (Class IIa and Class IIb). There are 107 LBD gene members in Class I: melon (35, 32.7%), Arabidopsis (37, 34.5%) and cucumber (35, 32.7%). There are 15 LBD gene members in Class II: melon (5, 33.3%), Arabidopsis (6, 40%) and cucumber (4, 26.7%). Among these sub-classes, the structural features of these seven clusters showed that cluster Class Ie was the largest cluster with 34 members (including 12 CmLBDs) while the smallest clusters were Class IIb with 6 members (including 2 CmLBDs).

Figure 1 Phylogenetic relationship of LBD proteins in C. melo, C. sativus and A. thaliana.

The different colors displays different subgroups of the CmLBD family. The green, blue and yellow color represent LBD genes from Cucumis melo, Cucumber sativus and Arabidopsis thaliana, respectively. Class Ia-Ie and Class I, Class IIa and Class IIb represents the seven sub-family groups.

Through phylogenetic analysis, these 40 CmLBD genes were grouped into seven clusters in Fig. 2A. The gene structure or exon–intron structures composition of 40 CmLBD genes was analyzed by GSDS v2.0 and represented in Fig. 2B. The number of introns ranged from 0 to 3. It was found that most of the genes (63%) contained 1 intron, whereas the CmLBD09 gene contained 3 introns. The longest intron in terms of sequence length was identified in the CmLBD38 gene. Also, 11 CmLBD genes didn’t use any intron regions among all clusters. The number of exons in CmLBDs ranged from one to four. It was determined that 12 CmLBDs contain one exons, 25 CmLBDs contain two exons and 2 CmLBDs contain three exons. The highest number of exons, four exons, was determined in the CmLBD09 gene. The longest exon in terms of sequence length was determined in CmLBD03 and CmLBD30 genes. According to gene structure research, several members of the same subclass exhibit different structural characteristics. For instance, CmLBD genes from subclass Ia can have between 0 and 3 introns. It is hypothesized that during evolution, members of subclass Ia may have undergone gene splicing or the insertion of gene fragments. A total of 10 conserved motifs were determined in the melon CmLBD TF gene family members (Figs. 2C, 3). Among the motifs detected, their length ranged from 6-50 amino acids, with 1-6 motifs per CmLBD gene. Three conserved motifs, such as motif 1, motif 2, and motif 3, were found in all of the CmLBD proteins, meaning that motif 1 ∼3 may play a significant role in the CmLBD regulating melon development.

Figure 2 The phylogenetic relationships, conserved motifs and gene structures of CmLBD proteins and CmLBD genes.

(A) Phylogenetic relationships of the melon LBD gene family. (B) Gene structures of LBD in melon. Blue boxes indicate 5′- and 3′-un. The melon LBD amino acid sequence was aligned by ClustalW. Motifs 1-10 are shown in different colored boxes. The scale bar indicates 50 aa.

Figure 3 Conserved motif symbols created using WebLogo program.

Chromosome localization and synteny analysis of the melon CmLBD gene family

Chromosomal location analyses showed that 38 CmLBD genes were mapped into 12 chromosomes (Fig. 4). Among mapped genes, chromosome 11 (chr11) had the highest number since it contained six CmLBD genes. The following chromosomes were chromosome chr10, containing five CmLBD genes, the chr03, chr04 and chr12 chromosomes contained four CmLBD genes and the chr01 and chr06 chromosomes contained three CmLBD genes. Chr02 and chr08 chromosomes contained two CmLBD genes, and chr05 and chr07 contained only one gene. Moreover, CmLBD39 and CmLBD40 genes were not found on the reference melon chromosome database.

Figure 4 Chromosomal location of melon LBD genes.

Within the melon genome, chromosomes are represented by vertical bars and chromosome number is represented by numbers above each chromosome. The scale is in millions of bases (Mb) and indicates physical length.

To figure out the evolution of the CmLBD genes in the Cucurbitaceae family, syntenic relationships between C. melo, A. thaliana, C. sativus and C. melo were analyzed Synteny analysis showed that a large number of orthologous LBDs were found in C. melo compared with C. sativus and Arabidopsis (Fig. 5). The synteny analysis identified 20 LBD orthologous gene pairs of C. melo and A. thaliana and 48 pairs of C. melo and C. sativus (Fig. 5).

Figure 5 Synteny analysis of LBD genes in C. melo and two plant species.

The red lines highlight the syntenic LBD gene pairs, while gray lines in the background point out the collinear blocks within C. melo and other plant genomes.

The ratio between Ka/Ks values was calculated for CmLBD homologous gene pairs to evaluate the selective pressure throughout evolution. It was investigated that the Ka/Ks values of the CmLBD gene pairs were less than 1 in general. It suggests that during the evolutionary process, these genes should have passed robust purifying selection.

Analysis of cis-acting elements in melon

In CmLBD genes, the cis-acting elements were examined using PlantCARE tool. Three types of cis-acting elements were assigned in the promoter region of the 34 CmLBD genes. These included cis-acting elements related to abiotic stress, phytohormone responses, plant growth and development (Fig. 6). The highest number of elements was found in the plant development and growth response category that contained circadian (circadian control), RY-element (seed-specific regulation). Following, plant hormone response-related category which includes CGTCA-motif (methyl jasmonate (MeJA) responsiveness), TCA-element (salicylic acid signal response element), TGACG-motif (methyl jasmonate (MeJA) responsiveness), ABRE (abscisic acid signal response element) and finally abiotic stress responses category contained ARE (anaerobic regulatory element) and LTR (lowtemperature response) was found. In addition, it was detected that different types of cis-elements exist in the promoter regions of most CmLBD genes. Based on this perspective, it is possible to propose that these CmLBDs play a role in variou biological processes.

Figure 6 The cis-acting elements in promotors of CmLBD genes.

The different colors of the oval boxes represent different cis-acting elements in the 1.5 kb promoter region upstream of the CmLBD gene.

Gene expression and annotation of CmLBD genes

Six tissue transcriptome data of melon male flower, female flower, leaf, stem, ovary, and root tissues were used to detect the expression levels of LBD family genes. The analysis showed that among CmLBD genes 12 of them (CmLBD01, CmLBD04, CmLBD05, CmLBD06, CmLBD07, CmLBD016, CmLBD018, CmLBD019, CmLBD021, CmLBD024, CmLBD033, CmLBD036) were expressed in all tissues with different expression patterns. Two of the CmLBD genes (CmLBD08 and CmLBD029) expression was not detected in any tissue. CmLBD04 displayed a higher expression level in the male flower, female flower, leaf, stem and ovary tissues. On the other hand, CmLBD01, CmLBD18, and CmLBD26 genes were mainly expressed in root tissue, CmLBD19 and CmLBD28 genes were expressed in ovary tissue. While CmLBD06 and CmLBD16 genes were prominent in stem tissue, CmLBD23 and CmLBD37 gene was highly expressed in leaf tissue. In addition to the high expression level of the CmLBD25 gene in female flower tissue, the expression of CmLBD10, CmLBD14, and CmLBD35 genes in male flower tissue is remarkable according to the transcriptome data (Fig. 7).

Figure 7 Heatmap of the expression of CmLBD genes in six different tissues from the R (root), O (ovary), S (stem), L (leaf), FF (female flower) and MF (male flower) of plants according to the transcriptome data.

In silico analysis displayed that the number of differential gene (DEG) was 39 in total. The highest number was found in G vs P. In contrast, no significant DEG was detected in C vs P. Mainly, in total, the number of genes showing downregulation was more significant than the transcripts showing upregulation. In C vs G, 11 genes were up-regulated though two were down-regulated (Table S3). Also, in C vs R comparison, two genes were found to be down-regulated, and one gene was found to be up-regulated. In G vs P, 10 genes were found to be down-regulated though two were up-regulated. While 7 CmLBD genes were down-regulated and one CmLBD gene was up-regulated in G vs R, two genes were down-regulated, and three genes were up-regulated in P vs R comparison. These results show that the expression levels of the genes change during different developmental stages of the plant and that LBD genes play a crucial role in plant growth and developmental processes.

GO analysis was performed to describe the functions of CmLBD genes in the biological processes level in different tissue melons. The analysis indicated that most of the CmLBD genes involved in regulation (26.25%), developmental process (21.96%) and metabolic process (16.71%), respectively (Fig. 8A). In developmental process, mainly the organ development and morphogenesis processes were more prominent. Also, the GO analysis showed that CmLBD genes at different developmental stages in melon, biological processes were mostly in regulation (25%), developmental process (19.44%), and metabolic process (19.44%) (Fig. 8B).

Figure 8 Gene ontology biological process results of CmLBD genes by Blast2Go program (A) CmLBD genes in biological processes level in different tissue melon, (B) CmLBD genes in biological processes at different developmental stages in melon.

To confirm the transcriptome data previously uploaded to NCBI, among the analyzed genes, five of them were selected (CmLBD01, CmLBD03, CmLBD14, CmLBD16, and CmLBD18) according to the expression patterns. qRT-PCR was performed to analyze the expression of selected genes in different tissues (leaf, root, stem, female flower, male flower, and ovary). The results were mostly consistent with the transcriptome data (Fig. 9). It was detected that the five CmLBD genes are differentially expressed in the different tissues and play essential roles in melon tissue development.

Figure 9 qRT–PCR analysis of CmLBD genes in different melon tissues.

Discussion

Melon scientifically known as Cucumis melo is a highly valuable crop cultivated worldwide. It belongs to the Cucurbitaceae family and is an annual plant with diploid genetic makeup. TFs are proteins that regulate gene expression by specifically binding to cis-acting elements in the promoter regions of target genes. LBD TFs are involved in the regulation of plant growth, lateral organ morphogenesis, border formation, stress response and secondary metabolism in plants (Li et al., 2017a; Li et al., 2017b). The LBD gene family has a wide distribution in the plant kingdom and has been studied and identified in many plants (Yang, Yu & Wu, 2006; Cao et al., 2016; Liu et al., 2019; Xu et al., 2021; Huang et al., 2021; Tian et al., 2022). However, there are no studies on the LBD gene family in melon. In this study, CmLBD genes were identified and characterized for the first time using genome-wide analysis in melon. Other studies identified 43 in A. thaliana, 39 in cucumber, 42 in grapes, 44 in maize, 46 in tomato and 55 in Eucalyptus grandis (Wang et al., 2013a; Wu et al., 2014; Kong et al., 2017; Lu et al., 2018).

A phylogenetic tree was generated to illuminate the evolutionary relationships between CmLBD genes in different plant species. According to the phylogenetic tree that was constructed, a close evolutionary relationship was detected between cucumber and Arabidopsis. This suggests that CmLBD genes are highly conserved throughout the evolutionary process (Yang, Yu & Wu, 2006; Wu et al., 2014). It was found that CmLBD proteins falling in the same cluster have similar conserved motifs. The proteins appearing in the same cluster might have similar functions. CmLBD gene structure and protein conserved motif analysis showed that highly associated gene members tend to display similar motif structure, and exon/intron structure, as observed in other plants for example A. thaliana, Ginkgo, Brassica napus, Passiflora edulis. Furthermore, CmLBD gene structure analysis demonstrated that 11 of the 40 CmLBD genes contained zero introns, while 29 of the 40 CmLBD genes included different introns ranging from one to three. Previous studies have confirmed that many LBD genes in plants are intronless (Yang, Yu & Wu, 2006; Tian et al., 2022; Xie et al., 2020; Liang et al., 2022).

According to chromosome location analysis, 38 CmLBD genes were distributed on 12 chromosomes. CmLBD14 and CmLBD18 were distributed on chr05 and chr07, respectively, while the other CmLBD genes were distributed on chromosome 10. Throughout evolution, members of the LBD gene family have generated six homologous gene pairs by tandem gene duplications, with gene distributions similar to those previously reported in the model plants A. thaliana and cucumber. To investigate the evolutionary mechanisms of CmLBD s, collinearity analysis was applied to C. melo, C. sativus and A. thaliana. Synteny blocks between C. melo and C. sativus of LBD genes were significantly greater than those between C. melo and A. thaliana. These results proposed that the sequence of LBD genes may be conserved in the Cucurbitaceae family. The results are consistent with the results of LBD genes in different plants (Xie et al., 2020; Wang et al., 2021; Jin et al., 2022). TFs interact with cis-acting elements to activate genes and regulate the transcription of multiple genes (Xu, Luo & Hochholdinger, 2016). This study found that LBD promoters contain various motifs concerned with plant developmental stage, stress response and hormone regulation. Abscisic acid (ABA) responsiveness elements were widely distributed in CmLBD s. It was reported that a well-known anti-stress plant hormone that regulates many developmental processes throughout all stages of lateral root growth, the AtLBD14, gene was down-regulated by ABA (Xu, Luo & Hochholdinger, 2016; Jeon & Kim, 2018).

According to the tissue expression pattern, it was detected that several members of the CmLBD gene members were specifically expressed in all tissues. Based on the tissue expression pattern, some precious CmLBD genes might have functions in specific physiological processes. For example, CmLBD01, CmLBD18 and CmLBD26 genes were mainly expressed in root tissue and these genes were mainly expressed in root. Many studies reported that in B. napus, BnLBD46/120 and BnLBD15/104 were significantly expressed in root tissues, and their ortholog genes AtLBD37 and AtLBD38 were highly expressed in root tissues in A. thaliana (Rubin et al., 2009; Klepikova et al., 2016). CmLBD19, CmLBD28, CmLBD25, CmLBD10, CmLBD14, and CmLBD3 genes were prominent in different parts of the floral tissue. These results suggest that these genes have a comparatively conserved modulatory role in the course of the flower development process. In a previous study conducted with P. edulis, it was reported that PeLBD14, PeLBD23 and PeLBD25 genes were highly expressed in flower tissue (Liang et al., 2022). The expression of CmLBD genes in melon was primarily occurred in the flowers and ovary, suggesting that they may be important in controlling the development of the floral organs and the early development of the fruit.In this study, CmLBD genes had different expression levels at six stages of fruit development. The number of genes showing down-regulation was greater than the transcripts showing up-regulation in all comparisons. It was determined that the number of differentially expressed genes and their expression levels decreased, especially in the growing - post-climacteric stage (C4 stage). These results suggest that CmLBD genes expressed at different growing stages might play an important role in fruit development and ripening as reported in other studies (Wang et al., 2013a; Zheng et al., 2016). GO annotation analysis was performed to investigate the functions of CmLBD genes further. Consistent with this study, LBD genes might participate in regulating plant root, stem, leaf, flower tissues, and lateral organ development (Semiarti et al., 2001; Thatcher et al., 2012; Cabrera et al., 2014).

QRT-PCR confirmed expression levels of five genes in six different tissues. The qRT-PCR results were consistent with the transcriptome sequencing results. Primarily, CmLBD01 was highly expressed only in root tissue, suggesting that the gene can be root-specific and CmLBD03 and CmLBD14 genes have a high expression and were found in male–female flower and ovary tissues. It may be interpreted that these genes have an essential contribution to flower development. Different studies have shown that the LBD gene family has different expression levels during plant tissues and organ development, and it plays a vital role in stress responses (Jin et al., 2022; Liang et al., 2022; Tian et al., 2022).

Conclusions

In conclusion, this is the first genome-wide identification and analysis of LBD TF genes in C. melo. In this research, 40 melon CmLBD TF genes were identified. Their physicochemical traits, phylogenetic relationships, gene structure and motif analysis, chromosomal distribution and expression pattern were investigated. The analysis of expression revealed that CmLBD genes have significant functions in various tissues and developmental stages of melons. The findings of this study will enhance our understanding of how LBD genes regulate different biological processes, including growth and development in melons. Additionally, these results may offer significant for further research on the functional characterization of the melon LBD gene family.

Supplemental Information

Supplemental Information 1 Primer sequences used for qRT-PCR.

Click here for additional data file.

Supplemental Information 2 The detailed information of CmLBD genes in Cucumis melo

Click here for additional data file.

Supplemental Information 3 Differentially expressed CmLBD genes at different developmental stages in melon

C stage (climacteric stage), G stage (growing stage), R stage (ripening stage), P stage (post-climacteric stage). Fold changes are given in log2-based numbers (↑ up-regulation, ↓ down-regulation).

Click here for additional data file.

Supplemental Information 4 Raw data for qRT-PCR.

Click here for additional data file.

Additional Information and Declarations

Competing Interests

Author Contributions

Data Deposition

The author declares that there are no competing interests.

Ebru Derelli Tufekci conceived and designed the experiments, performed the experiments, analyzed the data, prepared figures and/or tables, authored or reviewed drafts of the article, and approved the final draft.

The following information was supplied regarding data availability:

The raw data are available in the Figures and Supplementary File.

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
