# Peer review of "Genome-wide identification and analysis of Lateral Organ Boundaries Domain (LBD) transcription factor gene family in melon (Cucumis melo L.)"

_PeerJ, doi:10.7717/peerj.16020_

## Round 0.1 · original submission · Minor Revisions

We are pleased to inform you that the reviewers have expressed favorable comments regarding your submission. To move forward with the publication process, we kindly request that you revise your work to address all of the reviewers' comments. Your attention to these comments will greatly assist in ensuring the quality and success of your submission. Thank you for your efforts.

Reviewer 1 ·

Basic reporting

Here, the authors did a broad description of the Lateral Organ Boundaries Domain (LBD) transcription factor (TF) gene family using different bioinformatic tools and RT-qPCR analysis in different tissues. The work is really well written, and the results increase the knowledge of the area from which other important hypotheses can be derived. I only have minor comments:

-Lines 20 and 61: correct plant-specific
-Line 76: Do you mean “synthesize” instead of synthesis?
-Line 182: Please indicate the gene's name corresponding to 219 and 1089 bp.
-Lines 184-185: Do you mean the largest (or longest) and the smallest (or shortest) protein?
-Lines 189-191: You mention that you used 40 melon amino acid sequences of CmLBDs to build the phylogenetic tree. However, in your results, you indicate that “122 CmLBD genes could be classified into two major groups”. How can be this possible? Maybe you’re referring to the 122 total proteins (ATs, KGNs, and CmLBDs). Please clarify this point
-Figure 2: Do you mean “Blue boxes indicate 59- and 39-UTR”?
-Figure 7: Please indicate where these expression data come from.
-Figure 8: Please indicate what are we seeing in A and B.
-Lines 282-283: You indicate that your results were mostly consistent with the transcriptome data. Did you calculate the correlation coefficient? Or how can we compare both data? You could show this the Pearson correlation (R) or you could plot both data on the same graph so that trends can be observed.
-What relationship could you find between the expression patterns and the tissues where the genes are expressed with the phylogeny results? Could you observe any trend regarding this? Please discuss.

Experimental design

The methods are well described with sufficient detail.

Validity of the findings

The results are well supported and the conclusions are well stated.

Additional comments

NA

Reviewer 2 ·

Basic reporting

1. Figure 1 needs to be redrawn. It is suggested that an annotation be added to the upper right corner of the figure to explain what the different colored squares represent, rather than just explaining them in the titles and legends.
2. In your interpretation of the results, you seem to focus more on explaining what each figure is, while I am more interested in understanding what each result reflects. For instance, for Figure 2, apart from the general description (number, what was detected, percentages), could you elaborate more on the relationship between the structure (intron-exon structure) and the function of the LBD genes?

Experimental design

no comment

Validity of the findings

no comment

Additional comments

no comment

---

## Round 0.2 · Minor Revisions

Thank you for taking into account the technical comments provided by the reviewers. Attached herewith, please find a file containing suggestions for improving your contribution in terms of editorial, grammar, and style. Kindly review these suggestions and send us a revised version when it is convenient for you. At this stage, it is not necessary to provide a rebuttal letter.

---

## Round 0.3 · accepted · Accept

Thank you for your speedy reply; we are thrilled to inform you that your paper has been accepted for publication in PeerJ!